# Magnitude and determinants of drug-related problems among patients admitted to medical wards of southwestern Ethiopian hospitals: A multicenter prospective observational study

**Firomsa Bekele**[1]*, **Tesfaye Tsegaye**[1], **Efrem Negash**[2], **Ginenus Fekadu**[3,4]

1 Department of Pharmacy, College of Health Science, Mettu University, Mettu, Ethiopia, 2 Department of Public Health, College of Health Science, Mettu University, Mettu, Ethiopia, 3 Department of Pharmacy, Institute of Health Science, Wollega University, Nekemte, Ethiopia, 4 School of Pharmacy, The Chinese University of Hong Kong, Shatin, NT, Hong Kong

* firomsabekele21@gmail.com

**Data Availability Statement:** Due to ethical restrictions by the research review committee of College of Health Sciences, Mettu University, the

## Abstract

### Background

Drug-related problem (DRP) is an event involving drugs that can impact the patient's desired goal of therapy. In hospitalized patients, DRPs happen during the whole process of drug use such as during prescription, dispensing, administration, and follow-up of their treatment. Unrecognized and unresolved DRPs lead to significant drug-related morbidity and mortality. Several studies conducted in different hospitals and countries showed a high incidence of DRPs among hospitalized patients. Despite the available gaps, there were scanties of studies conducted on DRPs among patients admitted to medical wards in Ethiopia. Therefore, this study assessed the magnitude of drug-related problems and associated factors among patients admitted to the medical wards of selected Southwestern Ethiopian hospitals.

### Patients and methods

A multicenter prospective observational study was conducted at medical wards of Mettu Karl Hospital, Bedele General Hospital and Darimu General Hospital. Adult patients greater than 18 years who were admitted to the non-intensive care unit (ICU) of medical wards and with more than 48 h of length of stay were included. Identified DRPs were recorded and classified using the pharmaceutical care network Europe foundation classification system and adverse drug reaction was assessed using the Naranjo algorithm of adverse drug reaction probability scale. Hill-Bone Compliance to High Blood Pressure Therapy Scale was used to measure medication adherence. Multivariable logistic regression was used to analyze the associations between the dependent variable and independent variables.

data underlying this study is available only upon request. Interested, qualified researchers can access the data by requesting research coordinator, College of Health Sciences Mettu University, Mohammedamin Hajure (sikoado340@gmail.com) and the corresponding author, Firomsa Bekele (firomsabekele21@gmail.com).

**Funding:** The author(s) received no specific funding for this work.

**Competing interests:** The authors have declared that no competing interests exist.

**Abbreviations:** ADE, Adverse Drug Event; ADR, Adverse drug reaction; AIDS, AIDS: Acquired immune deficiency disease; ASA, Acetyl Salicylic Acid; AOR, Adjusted odds ratio; BGH, Bedele General Hospital; COR, crude odds ratio; DGH, Darimu General Hospital: DRP: Drug related problem; DTP, Drug therapy problem; GIT, Gastro Intestinal Drug; ICU, Intensive care unit; MKH, Mettu Karl Hospital; NSAID, Non-Steroidal Inflammatory Drug; SD, Standard Deviation; SPSS, statistical package for social sciences; USA, United States of America; USD, United State Dollar.

## Result

Of the 313 study participants, 178 (56.9%) were males. The prevalence of actual or potential DRPs among study participants taking at least a single drug was 212 (67.7%). About 125 (36.63%) patients had one or more co-morbid disease and the average duration of hospital stay of 7.14 ± 4.731 days. A total of 331 DRPs were identified with an average 1.06 DRP per patient. The three-leading categories of DRPs were unnecessary prescription of drugs 92 (27.79%), non-adherence (17.22%) and dose too high (16.92%). The most common drugs associated with DRPs were ceftriaxone (28.37%), cimetidine (14.88%), and diclofenac (14.42%). The area of residence (AOR = 2.550, 95CI%: 1.238–5.253, p = 0.011), hospital stay more than 7 days (AOR = 9.785, 95CI%: 4.668–20.511, p≤0.001), poly pharmacy (AOR = 3.229, 95CI%: 1.433–7.278, p = 0.005) were predictors of drug-related problem in multivariable logistic regression analysis.

## Conclusion

The magnitude of drug therapy problems among patients admitted to the medical wards of study settings was found to be high. Therefore, the clinical pharmacy services should be established in hospitals to tackle the DTPs in this area. Additionally, healthcare providers of hospitals also should create awareness for patients seeking care from health facilities of the importance of rational drug usage.

## Background

Drug therapy problem (DTP) is an event occurred during disease treatment that can have the probability to affect the desired treatment outcomes of patients, such as dose too low, dose too high, adverse drug reaction, need additional drug therapy, unnecessary drug therapy, non-adherence, and ineffective drug therapy [1,2]. Despite used for disease treatment and prevention, drugs are also responsible for drug-related problems (DRPs) [3]. In hospitalized and chronic care patients, drug-related problems (DTPs) happen during the whole process of drug use such as during prescription, dispensing, administration, and follow-up of their treatment [4].

Due to several reasons, hospitalized patients are receiving more medications than those treated as outpatients [5]. The rate of hospital admission due to DTP was estimated to be 5–10%, and the magnitude of patients discharged with DRP was about 22%. Among the seven categories of DTP, non-adherence, inappropriate indication, and ADR were resulted in emergency department visits in 28% of the patients [6].

The impact of DTPs among hospitalized patients includes decreased quality of life, prolonged hospital stays, increased healthcare budget, finally leading to death [7]. More people die of inappropriate drug treatment than breast cancer, AIDS, and traffic accidents all together [8]. Due to their training, pharmacists can play a key role in identifying these DTPs in resolving actual DTPs and preventing potential DTPs through careful pharmaceutical practices [9].

In USA and Australia about$177.4 and £100707 billion was spent to manage the consequences of DTPs annually, respectively [1,9]. A study done in Lagos State University Teaching Hospital, Nigeria, reported that approximately1.83 million naira (15,466.60 USD) was spent on managing all the patients admitted due to adverse drug reactions (ADRs). Therefore, it is possible to reduce the economic impacts of DRPs on patients, general population, and health care by tackling any DRP before its occurrence [3,7].

ADRs cause patients to lose confidence toward their physicians and seek self-treatment options, which consequently precipitate additional ADRs [10]. About 17% of heart failure (HF) patients experience adverse effects of their medication and 6.7% of ADR-related hospitalizations to cardiac units [11]. Around 5% of all hospital admissions are the result of an ADR, and around 10%–20% of inpatients will have at least one ADR during their hospital stay [10]. Therefore, ADR is the most common cause of patient morbidity, mortality, and increased healthcare costs [6,7,12].

The findings from England, Uganda, Brazil, and North India showed that the prevalence of ADR among admitted patients was 6.5%, 4.5%, 11.5%, and 22.6%, respectively [13–16]. The high rate of ADR is due to polypharmacy, adequate monitoring of drugs and irrational prescription of drugs by unauthorized professional [13,16]. A study conducted in Australian showed that about 1.3 million people were injured each year due to medication errors and adverse drug events (ADEs) in 16.6% of admissions, resulting in permanent disability in 13.7% and death in 4.9% patients [17].

Hospitalized patients are more likely to be exposed to poly pharmacy. Some DRPs exist at the time of admission to hospital, while others arise during hospital management. On average 2.6 DRPs occur per patient in internal medicine wards in Norway and the presence of DRPs increased approximately linearly with the number of drugs used [18]. Know a day's more than half of hypertensive patient's treatment was un-controlled despite different combination therapy was used. The condition is due to DRPs such as non-adherence, ADRs, ineffective drugs, under dose and overdose [19].

Therefore, several studies conducted in different hospitals and countries showed high incidence of DRPs among hospitalized patients. However, there was no DTP study conducted in medical wards of Ilubabor and Buno bedele zone and this study assessed DRPs and its predictors in medical ward of Mettu karl, Bedele, and Darimu hospitals.

## Patients and methods

### Study setting, design and study period

A multicenter prospective observational study was conducted in three hospitals found in the Ilu ababor and Buno bedele zones. The included hospitals are Mettu Karl Hospital (MKH), Bedele General Hospital (BGH) and Darimu General Hospital (DGH) during the study period of March 1, 2020 to June 1, 2020. Iluababor and Buno bedele zones are found in southwest Ethiopia located at 572 km and 423 km away from Addis Ababa, the capital city of Ethiopia, respectively.

### Study participants and eligibility criteria

Adult patients more than 18 years who were admitted to the non-intensive care unit (ICU) of medical wards and with more than 48 h of length of stay were included. Patients who refused to participate, re-admitted during the study period, and developed ADR due to genetic factors were excluded.

### Study variables and outcome endpoints

The DTP was the primary outcome. ADR was assessed using the Naranjo algorithm of the ADR probability scale [20]. Hill-Bone Compliance to High Blood Pressure Therapy Scale was used to measure medication adherence. Accordingly, the nine-item medication-taking sub-scale was employed. Each item is a four-point Likert type scale (none of the time, some of the time, most of the time and all of the time) [21,22]. The total scores on this subscale range from

9 to 36, with higher scores reflecting poorer adherence to drug therapy. The median split was used and dichotomized into two groups 1 = Adherent to the treatment and 0 = Non-adherent to the medication.

## Sample size and sampling technique

The single population proportion formula was used to calculate the required sample size by considering the following assumptions: Proportion of drug-related problems was 75.51% from else report [6], 95% confidence level, and 5% margin of error (the absolute level of precision).

$$n = \frac{(Z\alpha/2)^2 \; p \; (1-p)}{d^2}$$

$$z = 1.96$$

$$P = 75.51\% \; (0.7551) \text{ and } d = 0.05$$

$$n = \frac{(1.96)^2 (0.755) \; (0.245)}{(0.05)^2} = 284.24 \sim 284$$

$$\text{Where, } n = \text{sample size}$$

A 10% contingency yielded a final sample size of 313. Proportional allocation was used to select study subjects based on the number of patients that the respective hospitals contained in their medical wards. From the total sample size calculated, 190 patients were from MKH, 68 patients were from BGH and 55 patients were from DGH. The subjects were chosen using a consecutive sampling technique.

## Data collection process and management

Data were collected using questionnaire which was developed after reviewing different literature and checklist was prepared to verify the patient's medical information. Four medical doctors, four nurses and four pharmacists were recruited for data collection. Additionally, three medical doctors and three pharmacists were assigned to supervise the data collection process. The supervisor and principal investigator were closely following the data collection process at the spot. The principal investigator evaluated the appropriateness of medical therapy using various references like Medscape, up to date, Lexicom and Micromedex and different guidelines. Identified DRPs were recorded and classified using the DRP registration format [2]. ADR was assessed using Naranjo algorithm of ADR probability scale. Accordingly, ADR Probability scale was categorized by taking sum the of 10 questions and grouped as definite, probable, possible, or doubtful if the total score is $\geq 9$, 5–8, 1–4 and 0, respectively [20].

Five percent of the sample was pre-tested to check the acceptability and consistency of the data collection tool two weeks before the actual data collection.

## Data processing and analysis

EPI-Info 3.5.4 software was used to enter the data. The principal investigator checked and evaluated the completeness of data before conducting the analysis. Finally, statistical software for social sciences (SPSS) 23.0 was used to analyze the data. Descriptive data were placed as frequency and percentage. Results were expressed as proportions and as means ± Standard Deviations (SD) based on the type of data. Bivariate and multivariate logistic regression were used

to analyze the associations between the dependent variable and independent variables. A p-value of less than 0.05 was considered statistically significant.

### Ethics approval and consent-to-participate

Ethical clearance was obtained from the Research Ethics Review Committee (RERC) of Mettu University. Permission was obtained from the medical director of the MKH, DH and BH to access medical ward patients and conduct the study. The benefits and risks of the study were explained to each participant included in the study. Written and oral informed consent were obtained from each patient involved in the study. The approval of using oral informed consent was obtained from the Research Ethics Review Committee (RERC) of Mettu University. To ensure confidentiality, name and other identifiers of patients and healthcare professionals was not recorded on the data collection tools.

### Operational definitions

**Drug-related problem.** Includes ADR, non-adherence, inappropriate indication and dose, drug interaction and ineffective drug therapy.

**Unnecessary drug therapy.** If fulfilled the following criteria [2].

- There is no valid medical indication for drug therapy at the time.

- Multiple drug products are being used for a condition that requires a single drug therapy.

- The medical condition is more appropriately treated with nondrug therapy.

- Drug therapy is being taken to treat an avoidable adverse reaction associated with another medication.

- Drug abuse, alcohol use, or smoking is causing the problem.

**Hospital stay.** The duration from patient's admission to discharge.

**Medication error.** Any preventable event that cause or lead to inappropriate medication use or patient harm while the medication is in the control of the healthcare professional, patient or consumer.

**Pharmaceutical care.** The process through which a pharmacist cooperates with a patient and other professional in designing, implementing, and monitoring a therapeutic plan that will produce specific therapeutic outcomes for the patient.

**Polypharmacy.** The daily consumption of 5 or more medications [7].

**Co-morbidity.** The presence of two or more diseases [11].

## Results

### Socio-demographic characteristics of study participants

Of the 313 eligible patients admitted to medical wards of MKH, BGH, and DGH about 178 (56.9%) were males and about 163 (52.1%) of them were in the age range of 18–35 years. More than 171(54.6%) were married and 179 (57.2%) were farmers by occupation (**Table 1**).

### Lifestyle and clinical characteristics of the study participants

A total of 125 (36.63%) patients had one or more co-morbid disease and 127 (59.4%) patients had a prolonged hospital stay (>7 days) with an average duration of hospital stay of 7.14 + 4.731 days. Regarding their lifestyle, 62 (19.8%) of them had habit of alcohol drink (**Table 2**).

**Table 1. Socio-demographic characteristics of patients admitted to medical wards of MKH, BGH, and DGH, 2020.**

| Variables | | Frequency (n) | Percentage (%) |
|---|---|---|---|
| Sex | Male | 178 | 56.9 |
| | Female | 135 | 43.1 |
| Age (years) | 18–35 | 163 | 52.1 |
| | 36–64 | 124 | 39.6 |
| | ≥ 65 | 26 | 8.3 |
| Marital status | Married | 171 | 54.6 |
| | Single | 95 | 30.4 |
| | Divorced | 31 | 9.9 |
| | Widowed | 16 | 5.1 |
| Educational status | illiterate | 61 | 19.5 |
| | Grade 1–8 | 114 | 36.4 |
| | Grade 9–12 | 60 | 19.2 |
| | Diploma | 43 | 13.7 |
| | Degree and above | 35 | 11.2 |
| Occupation | Farmer | 179 | 57.2 |
| | Merchant/trade | 18 | 5.8 |
| | Government employee | 53 | 16.9 |
| | Homemaker | 18 | 5.8 |
| | Student | 38 | 12.1 |
| | Others* | 7 | 2.2 |
| Residence | Urban | 70 | 22.4 |
| | Rural | 243 | 77.6 |

Others*: Daily laborer, non-governmental employee.

## Incidences of DRPs and common drugs involved in DRPs

The incidence of actual or potential DTPs among subjects who were taking at least a single drug was 212 (67.7%). A total of 331 DRPs were identified on average with 1.057 DRPs per patient. The three-leading category of drug-related problems found to be a culprit was unnecessary 92 (27.79%), non-adherence 57 (17.22%) and dose too high 56 (16.92%) (**Table 3**).

**Table 2. Lifestyle and clinical characteristics of patients admitted to medical wards of MKH, BGH, and DGH, 2020.**

| Variables | | Frequency (n) | Percentage (%) |
|---|---|---|---|
| Drink alcohol | Yes | 62 | 19.8 |
| | No | 251 | 80.2 |
| Chew a chat | Yes | 45 | 14.4 |
| | No | 268 | 85.6 |
| Smoke a cigarette | Yes | 25 | 8 |
| | No | 288 | 92 |
| Presence of comorbidity | Yes | 125 | 36.63 |
| | No | 188 | 60.1 |
| Length of hospital stay (days) | <7 | 186 | 39.9 |
| | ≥7 | 127 | 59.4 |
| Number of medications per patient | <5 | 241 | 77 |
| | ≥5 | 72 | 23 |

**Table 3. Types of drug therapy problems among patients admitted to the medical ward of MKRH, BH and DH.**

| Types of DRPs | Frequency (n) | Percentage (%) |
|---|---|---|
| Unnecessary drug therapy | 92 | 27.79% |
| Non-adherence | 57 | 17.22% |
| Dose too high | 56 | 16.92% |
| Needs additional drug therapy | 50 | 15.11% |
| Dose too low | 38 | 11.48% |
| Ineffective drug therapy | 30 | 9.06% |
| ADR | 8 | 2.42% |

Regarding the patients that were developed ADR, about 4 (50%), 3 (37.5%),1(12.5%) were definite, probable, possible ADR.

## The common drugs involved in DRPs

The most common drugs responsible for drug therapy problems were ceftriaxone 61(28.37%), cimetidine 32(14.88%), and diclofenac 31(14.42%) (Table 4).

Of the patients prescribed Ceftriaxone, 34(55.74%) of them were in the age range of 36–64 years. However, doxycycline was commonly given (45.83%) in the age range of 18–35 years. With regard to gender, cimetidine was commonly prescribed in females 18(56.25%) (Table 5).

## Factors associated with DRPs

The results of the multivariable logistic regression showed that there is a significant association between the area of residence, length of hospital stay, and polypharmacy with the presence of DRPs. Patients living in rural areas were 2.5 times more likely to have at least one DTP than patients who live in urban areas (AOR = 2.550,95CI%: 1.238–5.253, p = 0.011). Patients whose hospital stays greater than or equal to seven days were about 10 times more likely to have DRPs than patients whose hospital stay less than seven days (AOR = 9.785,95CI%: 4.668–20.511, p≤0.001). Lastly, patients who have prescribed 5 or more drugs (polypharmacy) were about 3 times more likely to have DRPs than patients prescribed with less than 5 drugs (AOR = 3.229,95CI%: 1.433–7.278, p = 0.005) (Table 6).

## Discussion

This study assessed factors associated with an increased risk of developing one or more DRP among patients admitted to medical wards in selected hospitals of South Western Ethiopia.

**Table 4. Common drugs associated with the occurrence of DRPs among patients admitted to the medical ward of MKRH, BGH and DGH, 2020.**

| Individual Drugs | Frequency (n) | Percentage (%) |
|---|---|---|
| Ceftriaxone | 61 | 28.37 |
| Cimetidine | 32 | 14.88 |
| Diclofenac | 31 | 14.42 |
| Furosemide | 26 | 12.09 |
| Doxycycline | 24 | 11.16 |
| Metronidazole | 20 | 9.30 |
| Tramadol | 13 | 6.05 |
| Augumentin | 12 | 5.58 |
| Acetyl salicylic acid | 10 | 4.65 |
| Spironolactone | 9 | 4.19 |

**Table 5. Major drugs associated with DRPs in terms of socio-demographic and morbidity characteristics among patients admitted to the medical ward of MKRH, BGH and DGH.**

| Variables Category | | Categories of Drugs | | | | |
|---|---|---|---|---|---|---|
| | | Ceftriaxone | Cimetidine | Diclofenac | Furosemide | Doxycycline |
| Age (years) | 18–35 | 18(29.51) | 7(21.88) | 10(32.26) | 9(34.62) | 11(45.83) |
| | 36–64 | 34(55.74) | 19(59.38) | 16(51.61) | 11(42.31) | 8(33.33) |
| | ≥ 65 | 9(14.75) | 6(18.75) | 5(16.13) | 6(23.08) | 5(20.83) |
| Sex | Male | 34(55.74) | 14(43.75) | 16(51.61) | 17(65.38) | 16(66.67) |
| | Female | 27(44.26) | 18(56.25) | 15(48.39) | 9(34.62) | 8(33.33) |
| Comorbidity | Yes | 40(65.57) | 19(59.38) | 20(64.52) | 15(57.69) | 15(62.50) |
| | No | 21(34.43) | 13(40.63) | 11(35.48) | 11(42.31) | 9(37.50) |
| Residence | Rural | 32(52.46) | 19(59.38) | 13(41.94) | 14(53.85) | 10(41.67) |
| | Urban | 29(47.54) | 13(40.63) | 18(58.06) | 12(46.15) | 14(58.33) |

The prevalence of DRP in the study area was 212 (67.7%), which complies with the study conducted in Gondar (66%) [7], Northern Sweden (66%) [23]. However, the magnitude of the DTP was lower than the study done in Dessie referral hospital (75.51%) [6], Kenya (93.8%) [24], Tikur Anbesa Specialized hospital (70.4%) [18], and Jimma University specialized hospital (73.5%) [11]. The magnitude was also higher than the study conducted in India in which the overall incidence of DRPs was found to be 47.66% [4].

The large variation seen on the magnitude of DRPs across studies might be due to different classification of systems used to classify DRPs and settings in which DRPs were assessed. Despite the difference seen in different areas, the magnitude of DRP is high that requires further intervention to tackle the progress and improve the patient's quality of life. This intervention requires collaborative efforts from the different stakeholders, patients and policy makers.

**Table 6. Multivariable logistic regression analysis results of factors associated with DRPs among patients admitted to Medical Wards of MKRH, BGH and DGH.**

| Variables | Category | DRPs | | COR (95%CI) | p-value | AOR (95%CI) | P-value |
|---|---|---|---|---|---|---|---|
| | | Yes (n = 212) | No (n = 101) | | | | |
| Sex | Male | 126 (59.43%) | 52(51.49%) | 1.38(0.857–2.22) * | 0.185 | 1.54(0.892–2.665) | 0.121 |
| | Female | 86(40.57%) | 49(48.51%) | 1 | | 1 | |
| Area of residence | Urban | 47(22.17%) | 23(22.77%) | 1 | | 1 | 0.011 |
| | Rural | 165(77.83%) | 78(77.23%) | 0.966 (0.548–1.703) * | 0.1905 | 2.550(1.238–5.253) ** | |
| Presence of comorbidity | Yes | 85(40.09%) | 40(39.60%) | 1.021 (0.629–1.66) * | 0.193 | 1.41(0.775–2.558) | 0.261 |
| | No | 127(59.91%) | 61(60.40%) | 1 | | 1 | |
| Poly pharmacy | Yes | 63(29.72%) | 9(8.91%) | 4.322 (2.051–9.106) * | <0.001 | 3.229 (1.433–7.278) ** | 0.005 |
| | No | 149(70.28%) | 92(91.09%) | 1 | | 1 | |
| Length of hospital stay (days) | ≥7 | 113(53.30%) | 14(13.86%) | 7.093 (3.794–13.259) * | <0.001 | 9.785 (4.668–20.511) ** | <0.001 |
| | <7 | 99(46.70%) | 87(86.14%) | 1 | | 1 | |
| Drink alcohol | Yes | 47(22.17%) | 15(14.85%) | 1.633(.864–3.088) * | 0.131 | 2.218(0.959–5.128) | 0.063 |
| | No | 165(77.83%) | 86(85.15%) | 1 | | 1 | |
| Smoke cigarette | Yes | 18(8.49%) | 7(6.93%) | 1.246(0.503–3.086) | * 0.164 | 0.445(0.139–1.429) | 0.174 |
| | No | 194(91.51%) | 94(93.07%) | 1 | | 1 | |
| Chew a chat | Yes | 25(11.79%) | 20(19.80%) | 0.541(0.285–1.030) * | 0.062 | 1.714(0.775–3.793) | 0.183 |
| | No | 187(88.21%) | 81(80.20%) | 1 | | 1 | |

*Shows significant at p-value 0.25

**Shows statistically significant at p-value 0.05.

AOR: Adjusted odds ratio; COR: Crudes odds ratio.

Furthermore, we recommend future studies to use similar DRP classification systems to generate comparable evidence.

In this study, unnecessary drug therapy 92 (27.79%), non-adherence 57 (17.22%), and dose too high 56 (16.92%) were most frequently encountered DRPs. The finding was consistent with the study of Dilla university referral hospital in which non-adherence was the most occurred types of DRP 68(29.69%), and ADR was the least type of DRP 7(3.06%) [25]. Additionally, the study at the University of Gondar reported inappropriate dose was the prevalent DRP [7] and in India, ADR was the least occurred type of DRP (1.39% [4]. However, non-adherence was the least common type of drug therapy problem in Jimma University Specialized Hospital [11]. Alternatively, ADR (18.6%) was the most occurred DRP in Adama hospital medical college [19].

In our study, the most common drugs associated with at least one of the drug therapy problems were ceftriaxone 61 (28.37%), cimetidine 32(14.88%), and diclofenac 31(14.42%). Similarly, a study conducted in Dessie referral hospital revealed that ceftriaxone (25.81%) was the most frequent specific drug for DTP and non-steroidal anti-inflammatory drugs accounts, 9 (4.84%) [6]. The study conducted in India showed that Antibiotics (33.5%), GIT drugs (29.3%) and NSAID (16.08%) were the most involved in DTPs [4]. In contrast, a study conducted at the university of Gondar showed that the most common agents associated with DTPs were omeprazole 45 (17.6%), heparin 22 (8.6%), and aspirin 21 (8.2%) [7]. A study conducted in Hong Kong revealed that Gentamicin, ranitidine and fluconazole were most frequently associated with DRPs [26]. The variety of drugs involved in DTPs was due to availability, patients and physician preferences, and different treatment guidelines within different countries.

In this study, 331 DRPs were recorded with 1.057 DRPs per patient. This finding is lower than study conducted among Hiwotfana specialized university hospital [27], Kenya [24], but comparable with the study of University of Gondar Teaching Hospital, Northwest Ethiopia [7]. The large variation seen on the magnitude of DRPs across studies might be due to a different classification system used to classify DRPs, settings in which DRPs were assessed, and differences in sample size of the study participants.

In this study, it was found the association between the risk of DRP and area of residence. Patients who live in rural areas were about 2.5 times more likely to develop drug-related problems compared to patients who were in urban area. The finding was inconsistent with the study of Spain in which, among demographic variables, only female sex was associated with a higher risk of developing at least one DRP [28]. Similarly, the area of residence was not a determinant of DRP in Jimma University specialized hospital [29]. The highest risk of DRP in rural areas was due to the lack of access to health information because of long distance to reach the health facilities.

Logistic regression analysis showed that the number of drugs prescribed per patient ($\geq 5$ prescribed drugs) was strong one predictor of the occurrence of DRP. The finding was consistent with the study in Hong Kong [26], Singapore [30], Dilla University Referral Hospital [25], and Jimma University Specialized Hospital [29]. The reason might be the potential drug-drug interaction as the results of polypharmacy.

In our study, patients who had a prolonged hospital stay were higher risk for developing at least one type of DTP. Our outcome was consistent with that of study-done university of Gondar [7]. In contrast, in Jimma's University Specialized Hospital, the length of hospital stay did not predict the presence of drug-related problems [11]. The reason might be the patients who had prolonged hospital stay have higher probability to develop different nosocomial infections requiring complex therapeutic management.

As the strength, the study was a multicenter and prospective as well as information on different organ function tests (renal, liver), diagnostic and laboratory tests were used to assess

any drug therapy problems. Our study has some limitations. First, the study was a cross-sectional study, which is difficult to determine the causal effect relationship. In addition, we only evaluate DTPs among patients admitted to the medical ward, which lacks generalizability. Finally, COVID-19 pandemic increased the stress of physicians and health workers to manage the patient's drug therapy.

## Conclusion

About two-thirds of the patients admitted to the medical ward of MKH, BGH, and DGH were experienced DRP. The three-leading category of drug-related problems found to be a culprit was unnecessary, non-adherence and dose too high. Ceftriaxone, cimetidine and diclofenac were the most common drugs involved in DTPs. The area of the residence, polypharmacy, and prolonged hospital stay were the predictors of the DTPs. Therefore, clinical pharmacy services should be established in hospitals to tackle any drug-related problems in our study area and healthcare providers of hospitals should create an awareness of patients seeking care from health facility about the importance of rational drug utilization. Policy makers and different stakeholders should also focus on the way to reduce the occurrence DTPs and improve the outcome. Future studies with large sample size at multiple hospitals for a longer period are highly warranted.

## Acknowledgments

We thank Mettu University for logistic support. We are grateful to staff members and healthcare professionals of Metu karl, Bedele, and Darimu hospital, data collectors, and study participants for their cooperation in the success of this study.

## Author Contributions

**Conceptualization:** Firomsa Bekele, Tesfaye Tsegaye.

**Data curation:** Firomsa Bekele, Tesfaye Tsegaye.

**Formal analysis:** Firomsa Bekele, Tesfaye Tsegaye, Ginenus Fekadu.

**Investigation:** Firomsa Bekele, Efrem Negash.

**Methodology:** Firomsa Bekele, Efrem Negash, Ginenus Fekadu.

**Visualization:** Tesfaye Tsegaye, Efrem Negash.

**Writing – original draft:** Firomsa Bekele.

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
