## [Decision Letter · Decision Letter 0]

8 Dec 2020

PONE-D-20-30167

Magnitude and Determinants of Drug Related Problems among Patients Admitted to Medical wards of Ilu Ababor and Buno bedele zonal Hospital: A Multicenter Prospective Observational Study

PLOS ONE

Dear Dr. Bekele,

Thank you for submitting your manuscript to PLOS ONE. After careful consideration, we feel that it has merit but does not fully meet PLOS ONE’s publication criteria as it currently stands. Therefore, we invite you to submit a revised version of the manuscript that addresses the points raised during the review process.

We look forward to receiving your revised manuscript.

Kind regards,

M. Mahmud Khan

Academic Editor

PLOS ONE

Additional Editor Comments:

In addition to the reviewer's report, I have reviewed the paper and my comments are below:

1. The paper should provide more information on drugs used by patient characteristics, i.e., major drugs with problems can be cross-tabulated with age, morbidity status, etc.

2. In the introduction, authors mentioned that a significant proportion of patients get admitted to hospitals due to adverse drug events. What approach did the study follow to ensure that the patient was not admitted due to adverse drug reactions?

3. Define the drug therapy problem "Unnecessary drug therapy" and since it is quite high, is there some specific patient groups that are more likely to be affected by this? Does it indicate error in physician prescription?

4. Paper needs thorough editing and revisions.

5. Methodology should be clearly presented (see the reviewer #1 comment on this as well)

Journal Requirements:

2. Thank you for including your ethics statement: "Ethical clearance was obtained from the ethics review board of Mettu University. Permission was obtained from medical director of the MKH, DH and BH to access medical ward patients and conducts the study."   

 a.Please amend your current ethics statement to include the full name of the ethics committee/institutional review board(s) that approved your specific study.

 b.Once you have amended this/these statement(s) in the Methods section of the manuscript, please add the same text to the “Ethics Statement” field of the submission form (via “Edit Submission”).

3. In the Methods, please state:

- Why written consent could not be obtained

- Whether the Institutional Review Board (IRB) approved use of oral consent

- How oral consent was documented

For more information, please see our guidelines for human subjects research: https://journals.plos.org/plosone/s/submission-guidelines#loc-human-subjects-research

4. We suggest you thoroughly copyedit your manuscript for language usage, spelling, and grammar. If you do not know anyone who can help you do this, you may wish to consider employing a professional scientific editing service.  

6. Thank you for submitting the above manuscript to PLOS ONE. During our internal evaluation of the manuscript, we found significant text overlap between your submission and the following previously published works, some of which you are an author.

- https://www.jbclinpharm.org/articles/assessment-of-drug-related-problems-and-its-associated-factors-among-medical-ward-patients-in-university-of-gondar-teaching-hospit.html

- https://link.springer.com/article/10.1007%2Fs11096-017-0504-9

- http://isfcppharmaspire.com/issueforpublication.aspx?Article=PHARMASPIRE_1_2017

- https://www.hindawi.com/journals/cdtp/2020/2509875/

- https://bmchealthservres.biomedcentral.com/articles/10.1186/s12913-018-3612-x

https://www.research.ed.ac.uk/portal/en/publications/the-apical-protein-apnoia-interacts-with-crumbs-to-regulate-tracheal-growth-and-inflation(70776d5d-4841-4fe5-b7f9-c614328149ee).html

- https://www.omicsonline.org/open-access/assessment-of-drug-related-problems-among-hypertensive-patients-2167-065X.1000122.php?aid=31481

The text that needs to be addressed involves the Background section of your manuscript.

Please revise the manuscript to rephrase the duplicated text, cite your sources, and provide details as to how the current manuscript advances on previous work. Please note that further consideration is dependent on the submission of a manuscript that addresses these concerns about the overlap in text with published work.

Reviewers' comments:

Reviewer's Responses to Questions

**Comments to the Author**

1. Is the manuscript technically sound, and do the data support the conclusions?

Reviewer #1: Partly

2. Has the statistical analysis been performed appropriately and rigorously? 

Reviewer #1: No

3. Have the authors made all data underlying the findings in their manuscript fully available?

Reviewer #1: Yes

4. Is the manuscript presented in an intelligible fashion and written in standard English?

Reviewer #1: No

5. Review Comments to the Author

Reviewer #1: The manuscript needs proof reading. there are awkward sentences that are difficult to understand that can be easily fixed at a proof reading stage. For example the first sentence in the abstract is not correct. Also in many places through the manuscript words that shouldn’t be capitalized are capitalized.

In many instances of importance to evaluate the manuscript rigor the provided text is missing critical information.

For example the author say in the method section, it says a questionnaire was developed. But then the authors reference medical chart review. The connection between the different steps need to be further developed to allow the reader to understand what was actually done.

6. PLOS authors have the option to publish the peer review history of their article (what does this mean?). If published, this will include your full peer review and any attached files.

Reviewer #1: No

---

## [Author Response · Author response to Decision Letter 0]

6 Jan 2021

M. Mahmud Khan

Editor, PLOS ONE

Dear Editor in chief of the Manuscript PONE-D-20-30167 entitled " Magnitude and Determinants of Drug Related Problems among Patients Admitted to Medical wards of Ilu Ababor and Buno bedele zonal Hospital: A Multicenter Prospective Observational Study," submitted to PLOS ONE. Thanks for your time and consideration in editing and reviewing the manuscript. We have carefully read your comments and corrected inline of your comments and suggestions. All comments raised were edited and incorporated in the revised manuscript. 

Here are the responses and elaborations for the comments from editor and reviewer!

EDITORS COMMENTS

Editor comment: The paper should provide more information on drugs used by patient characteristics, i.e., major drugs with problems can be cross-tabulated with age, morbidity status, etc.

Author response: In table 5 of revised manuscript most common drugs was cross tabulated with the variables like sex, residence, co-morbidity and age 

Editor comment: In the introduction, authors mentioned that a significant proportion of patients get admitted to hospitals due to adverse drug events. What approach did the study follow to ensure that the patient was not admitted due to adverse drug reactions?

Author response: ADR was assessed by using the Naranjo algorithm of the ADR probability scale

Editor comment: Define the drug therapy problem "Unnecessary drug therapy" and since it is quite high, is there some specific patient groups that are more likely to be affected by this? Does it indicate error in physician prescription?

Author response: In operational definition we have added the definition and criteria of unnecessary drug therapy

Editor comment: Paper needs thorough editing and revisions.

Author response: The whole manuscript was edited as per your comments

Editor comment: Methodology should be clearly presented (see the reviewer #1 comment on this as well)

Author response: We carefully revised and edited the methodology part

Editor comment: Regarding o Journal Requirements

Author response: The manuscript was written as per PLOS ONE's style

Editor comment: Please amend your current ethics statement to include the full name of the ethics committee/institutional review board(s) that approved your specific study.

Author response: The full name of the ethics committee/institutional review board(s) was written in revised manuscript and Ethics Statement” field of the submission

The written consent was be obtained

The statements showing that the Institutional Review Board (IRB) approved use of oral consent was written in revised manuscript

Editor comment: We suggest you thoroughly copyedit your manuscript for language usage, spelling, and grammar.

Author response: We have edited the English grammar throughout the manuscripts

 Editor comment: Regarding the data availability statement

Author response: We have updated the data availability statement in which it is placed as supporting information. 

Editor comment: Regarding to text overlap

Author response: The manuscript was paraphrased to minimize the texts overlap

REVIEWER COMMENTS

Reviewer 1 

Reviewer comments: The conclusions must be drawn appropriately based on the data presented.

Author response: The conclusion was edited to relate with data presented 

Reviewer comments: Regarding statistical analysis

Author response: The data was re-analyzed 

Reviewer comments: Is the manuscript presented in an intelligible fashion and written in Standard English?

Author response: We have edited the English grammar throughout the manuscripts

Reviewer comments: The manuscript needs proof reading. There are awkward sentences that are difficult to understand that can be easily fixed at a proof reading stage. For example the first sentence in the abstract is not correct. Also in many places through the manuscript words that shouldn’t be capitalized are capitalized.

Author response: The manuscript was edited carefully and the first sentence in the abstract was corrected. Finally any capitalization was corrected

Reviewer comments: In many instances of importance to evaluate the manuscript rigor the provided text is missing critical information. For example the author say in the method section, it says a questionnaire was developed. But then the authors reference medical chart review. The connection between the different steps need to be further developed to allow the reader to understand what was actually done.

Author response: It was written as data was collected using questionnaire which was developed after reviewing different literature and checklist was prepared to verify the patient’s medical information in revised manuscript

Reviewer comments: Don’t capitalize the P in pharmaceutical 

Author response: The capitalization was removed

Reviewer comments: Replace DTP by DRP

Author response: In abstract under result DTP was replaced by DRP

Reviewer comments: Re word A total of 331 DRPs were identified on average, 1.057 DRPs per

Patient to a total of 331 DRPs were identified, an average of 1.06 DRPs per patient.

Author response: It was written as per your comment

Reviewer comments: This sentence doesn’t make sense. Maybe something like the three-leading categories of DRPs were found to be unnecessary (not sure unnecessary what prescription of drug or consumption of drugs) with 92 cases (27.79%), non-adherence with 57 patients (17.22%) and high drug dosage with 56 cases (16.92%). 

Author response: It was corrected as prescribed drugs

Reviewer comments: Is the term drug therapy problem the same as DRPs?

Author response: We have used them interchangeably

Reviewer comments: missing reference on ADR and The connection between these two sentences is not clear.Also two sentences can’t make a paragraph you need at least 5. I think the authors are trying to convey the significance of the problem and that if 5-20% patients may lose trust in the physicians treating them. But the authors never explain why that is a bad thing or make the actual connection. 

Author response: The sentence regarding ADR was cited 

• The two sentence was connected appropriately and many references was added to explain more about DRP

• We have discussed the reasons for high burden of ADR

Reviewer comments: What treatment that was poorly controlled? This thought is not complete

Author response: It was corrected in revised manuscript as hypertensive patient’s was poorly controlled

Reviewer comments: This time frame in of concern since the pandemic really started. Wouldn’t that impact the stress of physicians and health workers. Should be addressed in the limitaitons

Author response: The impact of COVID-19 on stress of physicians was written as limitation 

Reviewer comments: You never explain ADR

Author response: ADR was assessed by using Naranjo algorithm of ADR probability scale. Accordingly, ADR Probability Scale was categorized by taking sum of 10 questions and grouped as definite, probable, possible or doubtful, if the total score is ≥9, 5-8, 1-4 and 0 respectively.

 Thanks for your time and consideration,

 Regards!

---

## [Editor Report · Decision Letter 1]

25 Jan 2021

PONE-D-20-30167R1

Magnitude and determinants of drug related problems among patients admitted to medical wards of south-western Ethiopian hospitals : A multicenter prospective observational Study

PLOS ONE

Dear Dr. Bekele,

Thank you for submitting your manuscript to PLOS ONE. After careful consideration, we feel that it has merit but does not fully meet PLOS ONE’s publication criteria as it currently stands. Therefore, we invite you to submit a revised version of the manuscript that addresses the points raised during the review process.

We look forward to receiving your revised manuscript.

Kind regards,

M. Mahmud Khan

Academic Editor

PLOS ONE

Additional Editor Comments (if provided):

Unfortunately, the paper needs thorough editing, maybe by a professional English language editor. I still find many language and expression related issues in the revised version. Without thorough editing, the paper cannot be accepted for publication.

Thanks

---

## [Author Response · Author response to Decision Letter 1]

8 Feb 2021

M. Mahmud Khan

Editor, PLOS ONE

Dear Editor in chief of the Manuscript PONE-D-20-30167 entitled " Magnitude and determinants of drug-related problems among patients admitted to medical wards of south-western Ethiopian hospitals: A multicenter prospective observational study" submitted to PLOS ONE. Thanks for your time and consideration in editing and reviewing the manuscript. We have carefully read your comments and corrected inline of your comments and suggestions. All comments raised were edited and incorporated in the revised manuscript. 

Here are the responses and elaborations for the comments from editor and reviewer!

EDITORS COMMENTS

Editor comment: The paper should provide more information on drugs used by patient characteristics, i.e., major drugs with problems can be cross-tabulated with age, morbidity status, etc.

Author response: In table 5 of revised manuscript most common drugs was cross tabulated with the variables like sex, residence, co-morbidity and age 

Editor comment: In the introduction, authors mentioned that a significant proportion of patients get admitted to hospitals due to adverse drug events. What approach did the study follow to ensure that the patient was not admitted due to adverse drug reactions?

Author response: ADR was assessed by using the Naranjo algorithm of the ADR probability scale

Editor comment: Define the drug therapy problem "Unnecessary drug therapy" and since it is quite high, is there some specific patient groups that are more likely to be affected by this? Does it indicate error in physician prescription?

Author response: In operational definition we have added the definition and criteria of unnecessary drug therapy. Patients presents with infectious disease were highly affected by unnecessary drug therapy because in our study are ceftriaxone was used unnecessarily. Any medication error occurred during prescribing, administration and dispensing was considered as unnecessary drug therapy

Editor comment: Paper needs thorough editing and revisions.

Author response: The whole manuscript was edited as per your comments

Editor comment: Methodology should be clearly presented (see the reviewer #1 comment on this as well)

Author response: We carefully revised and edited the methodology part

Editor comment: Regarding o Journal Requirements

Author response: The manuscript was written as per PLOS ONE's style

Editor comment: Please amend your current ethics statement to include the full name of the ethics committee/institutional review board(s) that approved your specific study.

Author response: The full name of the ethics committee/institutional review board(s) was written in revised manuscript and Ethics Statement” field of the submission

The written consent was be obtained

The statements showing that the Institutional Review Board (IRB) approved use of oral consent was written in revised manuscript

The oral consents was clearly documented in separate checklists

Editor comment: We suggest you thoroughly copyedit your manuscript for language usage, spelling, and grammar.

Author response: We have edited the English grammar throughout the manuscripts

 Editor comment: Regarding the data availability statement

Author response: We have updated the data availability statement in which it is restricted 

Editor comment: Regarding to text overlap

Author response: The manuscript was paraphrased to minimize the texts overlap

REVIEWER COMMENTS

Reviewer 1 

Reviewer comments: The conclusions must be drawn appropriately based on the data presented.

Author response: The conclusion was edited to relate with data presented 

Reviewer comments: Regarding statistical analysis

Author response: The data was re-analyzed 

Reviewer comments: Is the manuscript presented in an intelligible fashion and written in Standard English?

Author response: We have edited the English grammar throughout the manuscripts

Reviewer comments: The manuscript needs proof reading. There are awkward sentences that are difficult to understand that can be easily fixed at a proof reading stage. For example the first sentence in the abstract is not correct. Also in many places through the manuscript words that shouldn’t be capitalized are capitalized.

Author response: The manuscript was edited carefully and the first sentence in the abstract was corrected. Finally any capitalization was corrected

Reviewer comments: In many instances of importance to evaluate the manuscript rigor the provided text is missing critical information. For example the author say in the method section, it says a questionnaire was developed. But then the authors reference medical chart review. The connection between the different steps need to be further developed to allow the reader to understand what was actually done.

Author response: It was written as data was collected using questionnaire which was developed after reviewing different literature and checklist was prepared to verify the patient’s medical information in revised manuscript

Reviewer comments: Don’t capitalize the P in pharmaceutical 

Author response: The capitalization was removed

Reviewer comments: Replace DTP by DRP

Author response: In abstract under result DTP was replaced by DRP

Reviewer comments: Re word A total of 331 DRPs were identified on average, 1.057 DRPs per

Patient to a total of 331 DRPs were identified, an average of 1.06 DRPs per patient.

Author response: It was written as per your comment

Reviewer comments: This sentence doesn’t make sense. Maybe something like the three-leading categories of DRPs were found to be unnecessary (not sure unnecessary what prescription of drug or consumption of drugs) with 92 cases (27.79%), non-adherence with 57 patients (17.22%) and high drug dosage with 56 cases (16.92%). 

Author response: It was corrected as prescribed drugs

Reviewer comments: Is the term drug therapy problem the same as DRPs?

Author response: We have used them interchangeably

Reviewer comments: missing reference on ADR and The connection between these two sentences is not clear.Also two sentences can’t make a paragraph you need at least 5. I think the authors are trying to convey the significance of the problem and that if 5-20% patients may lose trust in the physicians treating them. But the authors never explain why that is a bad thing or make the actual connection. 

Author response: The sentence regarding ADR was cited 

• The two sentence was connected appropriately and many references was added to explain more about DRP

• We have discussed the reasons for high burden of ADR

Reviewer comments: What treatment that was poorly controlled? This thought is not complete

Author response: It was corrected in revised manuscript as hypertensive patient’s was poorly controlled

Reviewer comments: This time frame in of concern since the pandemic really started. Wouldn’t that impact the stress of physicians and health workers. Should be addressed in the limitaitons

Author response: The impact of COVID-19 on stress of physicians was written as limitation 

Reviewer comments: You never explain ADR

Author response: ADR was assessed by using Naranjo algorithm of ADR probability scale. Accordingly, ADR Probability Scale was categorized by taking sum of 10 questions and grouped as definite, probable, possible or doubtful, if the total score is ≥9, 5-8, 1-4 and 0 respectively.It is mentioned as ”Regarding to the patients that were developed ADR, about 4(50%), 3(37.5%),1(12.5%) were definite, probable, possible ADR’’.

 Thanks for your time and consideration,

 Regards!

---

## [Editor Report · Decision Letter 2]

2 Mar 2021

Magnitude and determinants of drug-related problems among patients admitted to medical wards of southwestern Ethiopian hospitals : A multicenter prospective observational study

PONE-D-20-30167R2

Dear Dr. Bekele,

We’re pleased to inform you that your manuscript has been judged scientifically suitable for publication and will be formally accepted for publication once it meets all outstanding technical requirements.

Kind regards,

M. Mahmud Khan

Academic Editor

PLOS ONE
---

## [Editor Report · Acceptance letter]

4 Mar 2021

PONE-D-20-30167R2 

Magnitude and determinants of drug-related problems among patients admitted to medical wards of southwestern Ethiopian hospitals: A multicenter prospective observational study 

Dear Dr. Bekele:

I'm pleased to inform you that your manuscript has been deemed suitable for publication in PLOS ONE. Congratulations! Your manuscript is now with our production department. 

Kind regards, 

on behalf of

Dr. M. Mahmud Khan 

Academic Editor

PLOS ONE